# VARIATION NETWORK: LEARNING HIGH-LEVEL ATTRIBUTES FOR CONTROLLED INPUT MANIPULATION

## ABSTRACT

This paper presents the *Variation Network* (VarNet), a generative model providing means to manipulate the high-level attributes of a given input. The originality of our approach is that VarNet is not only capable of handling pre-defined attributes but can also learn the relevant attributes of the dataset by itself. These two settings can be easily combined which makes VarNet applicable for a wide variety of tasks. Further, VarNet has a sound probabilistic interpretation which grants us with a novel way to navigate in the latent spaces as well as means to control how the attributes are learned. We demonstrate experimentally that this model is capable of performing interesting input manipulation and that the learned attributes are relevant and interpretable.

## 1 INTRODUCTION

We focus on the problem of generating *variations* of a given input in an intended way. This means that given some input element $x$, which can be considered as a *template*, we want to generate transformed versions of $x$ with different high-level *attributes*. Such a mechanism is of great use in many domains such as image edition since it allows to edit images on a more abstract level and is of crucial importance for creative uses since it allows to generate new content.

More precisely, given a dataset $\mathcal{D} = \{(x^{(1)}, m^{(1)}), \ldots, (x^{(N)}, m^{(N)})\}$ of $N$ labeled elements $(x, m) \in \mathcal{X} \times \mathcal{M}$, where $\mathcal{X}$ stands for the input space and $\mathcal{M}$ for the metadata space, we would like to obtain a model capable of learning a relevant *attribute space* $\Psi \subset \mathbf{R}^d$ for some integer $d > 0$ and meaningful *attribute functions* $\phi : \mathcal{X} \times \mathcal{M} \to \Psi$ that we can then use to control generation.

In a great majority of the recent proposed methods Lample et al. (2017); Upchurch et al. (2016), these attributes are assumed to be given. We identify two shortcomings: labeled data is not always available and this approach *de facto* excludes attributes that can be hard to formulate in an absolute way. The novelty of our approach is that these attributes can be either learned by the model (we name them *free attributes*) or imposed (*fixed attributes*). This problem is an ill-posed one on many aspects. Firstly, in the case of fixed attribute functions $\phi$, there is no ground truth for variations since there is no $x$ with two different attributes. Secondly, it can be hard to determine if a learned free attribute is relevant. However, we provide empirical evidence that our general approach is capable of learning such relevant attributes and that they can be used for generating meaningful variations.

In this paper, we introduce the *Variation Network* (VarNet), a probabilistic neural network which provides means to manipulate an input by changing its high-level attributes. Our model has a sound probabilistic interpretation which makes the variations obtained by changing the attributes statistically meaningful. As a consequence, this probabilistic framework provides us with a novel mechanism to "control" or "shape" the learned free attributes which then gives interpretable controls over the variations. This architecture is general and provides a wide range of choices for the design of the attribute function $\phi$: we can combine both free and fixed attributes and the fixed attributes can be either continuous or discrete.

Our contributions are the following:

- A widely applicable encoder-decoder architecture which generalizes existing approaches Kingma & Welling (2013); Rubenstein et al. (2018); Lample et al. (2017)

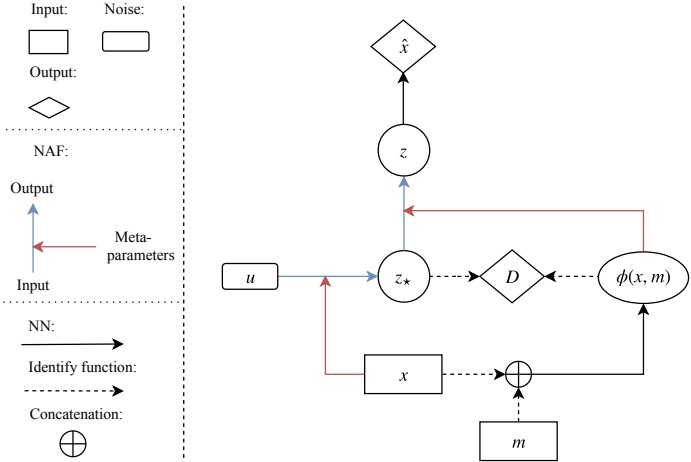

Figure 1: VarNet architecture. The input $x, \hat{x}$ are in $\mathcal{X}$, the input space and the metadata $m$ is in $\mathcal{M}$, the metadata space. The latent template code $z^*$ lies in $Z^*$, the template space, while the latent variable $z$ lies in $\mathcal{Z}$ the latent space. The variable $u$ is sampled from a zero-mean unit-variance normal distribution. Finally, the features $\phi(x, m)$ are in $\Psi$, the attribute space. The Neural Autoregressive Flows (NAF) Huang et al. (2018) are represented using two arrows, one pointing to the center of the other one; this denotes the fact that the actual parameters of first neural network are obtained by feeding meta-parameters into a second neural network. The discriminator $D$ acts on $\mathcal{Z}^* \times \Psi$.

- An easy-to-use framework: any encoder-decoder architecture can be easily transformed into a VarNet in order to provide it with controlled input manipulation capabilities,

- A novel and statistically sound approach to navigate in the latent space,

- Ways to control the behavior of the free learned attributes.

The plan of this paper is the following: Sect. 2 presents the VarNet architecture together with its training algorithm. For better clarity, we introduce separately all the components featured in our model and postpone the discussion about their interplay and the motivation behind our modeling choices in Sect. 3 and Sect. 4 discusses about the related works. In particular, we show that VarNet provides an interesting solution to many constrained generation problems already considered in the literature. Finally, we illustrate in Appendix A the possibilities offered by our proposed model and show that its faculty to generate variations in an intended way is of particular interest.

## 2 PROPOSED MODEL

We now introduce our novel encoder-decoder architecture which we name *Variation Network*. Our architecture borrows principles from the traditional Variational AutoEncoder (VAE) architecture Kingma & Welling (2013) and from the Wasserstein AutoEncoder (WAE) architecture Tolstikhin et al. (2017); Rubenstein et al. (2018). It uses an adversarially learned regularization Dumoulin et al. (2016); Lample et al. (2017), introduces a separate latent space for templates Adel et al. (2018) and decomposes the attributes on an adaptive basis Wang et al. (2018). It can be seen as a VAE with a particular decoder network or as a WAE with a particular encoder network. Our architecture is shown in Fig. 1 and our training algorithm is presented in Alg. 1.

We detail in the following sections the different parts involved in our model. In Sect. 2.1, we focus on the encoder-decoder part of VarNet and explain Eq. (3), (4) and (5). In Sect. 2.2, we introduce the adversarially-learned regularization whose aim is to disentangle attributes from templates (Eq. (1) and (6)). Section 2.3 discusses the special parametrization that we adopted for the attribute space $\Psi$.

---

**Algorithm 1** Variation Network training procedure

---

**Require:** Dataset $\mathcal{D} = \left\{ (x^{(i)}, m^{(i)} \right\}_{i=1..N}$, reconstruction cost $c$,
    reproducing kernel $k$, batch size $n$

1: **for** Fixed number of iterations **do**
2:    Sample $x := (x_1, \ldots, x_n)$ and $m := (m_1, \ldots, m_n)$ where $(x_i, m_i)$ i.i.d. samples from $\mathcal{D}$
3:    Sample $z_i^* \sim q^*(\cdot|x_i)$
4:    Compute $z := \{z_1, \ldots, z_n\}$ where $z_i = f_{\phi(x_i, m_i)}(z_i^*)$
5:    Sample $\hat{x} := \{\hat{x}_1, \ldots, \hat{x}_n\}$ where $\hat{x}_i \sim p(\cdot|z_i)$,
6:    Sample random features $\{\psi_i\}_{i=1..n}$ from feature space $\Psi$ using $\nu$ (see Sect. 2.3)
7:    Let $\tilde{z} := \{\tilde{z}_1, \ldots, \tilde{z}_n\}$ where $\tilde{z}_i \sim p(\cdot)$
8:    *Discriminator training phase*
9:    Compute

$$\mathcal{L}_{\text{Disc},n} := \frac{1}{n} \sum_{i=1}^{n} \log D\left(z_i^*, \psi_i\right) + \log\left(1 - D(z_i^*, \phi(x_i, m_i))\right) \tag{1}$$

10:   Gradient ascent step on the discriminator parameters using $\nabla \mathcal{L}_{\text{Disc}}$
11:   *Encoder-decoder training phase*
12:   Compute

$$\mathcal{L}_{\text{EncDec},n} := \text{RE}_n + \beta \text{KL}_n^* + \lambda \text{MMD}_{k,n} + \gamma \mathcal{R}_{\text{Disc},n} \tag{2}$$

    where

$$\text{RE}_n := \frac{1}{n} \sum_{i=1}^{n} c(x_i, \hat{x}_i), \tag{3}$$

$$\text{KL}_n^* := \frac{1}{n} \sum_{i=1}^{n} \log q^*(z_i^*|x_i) - \log p^*(z_i^*), \tag{4}$$

$$\text{MMD}_{k,n} := \frac{1}{n(n-1)} \sum_{l \neq k} k(z_l, z_j) + \frac{1}{n(n-1)} \sum_{l \neq k} k(\tilde{z}_l, \tilde{z}_j) - \frac{2}{n^2} \sum_{l,j} k(z_l, \tilde{z}_j), \tag{5}$$

$$\mathcal{R}_{\text{Disc},n} = -\frac{1}{n} \sum_{i=1}^{n} \log D(z_i^*, \phi(x_i, m_i)). \tag{6}$$

13:   Gradient ascent step on all parameters except the discriminator parameters (encoder and decoder parameters, feature function parameters, features vectors and NAF $f$) using $\nabla \mathcal{L}_{\text{EncDec}}$
14: **end for**

---

## 2.1 ENCODER-DECODER PART

Similar to the VAE architectures, we suppose that our data $x \in \mathcal{X}$ depends on some latent variable $z \in \mathcal{Z}$ through some decoder $p(x|z)$ parametrized by a neural network. We introduce a prior $p(z)$ over this latent space so that the joint probability distribution is expressed as $p(x, z) = p(x|z)p(z)$. Since the posterior distribution $p(z|x)$ is usually intractable, an approximate posterior distribution $q(z|x)$ parametrized by a neural network is usually introduced.

The novelty of our approach is on how we write this encoder network. Firstly, we introduce an attribute space $\Psi \subset \mathbf{R}^d$, where $d$ is the dimension of the attribute space, on which we condition the encoder which we now denote as $q(\cdot|x, \psi \in \Psi)$. More details about the attribute space $\Psi$ are given in Sect. 2.3. For the moment, we can consider it to be a subspace of $\mathbf{R}^d$ from which we can sample from. The objective in doing so is that decoding $z \sim q(\cdot|x, \psi)$ using $p(x|z)$ will result in a sample $\tilde{x}$ that is a variation of $x$ but with features $\psi$. Secondly, in order to correctly reconstruct $x$, introduce an *attribute function* $\phi : \mathcal{X} \times \mathcal{M} \to \Psi$ computed from $x$ and its metadata $m$ with values in the attribute space $\Psi$. This attribute function is a deterministic neural network that will be learned during training and whose aim is to compute attributes of $x$.

For an input $(x, m) \in \mathcal{D}$, we want to decouple a *template* obtained from $x$ from its attributes $\phi(x, m)$ computed from $x$ and (possibly) from its metadata $m$. This is done by introducing another latent space $\mathcal{Z}^*$ that we term *template space* together with a approximated posterior distribution $q^*(z^*|x)$

parametrized by a neural network and a fixed prior $p^*(z^*)$. The idea is then to compute $z$ from $z^*$ by applying a transformation parametrized *only* by the feature space $\Psi$. In practice, this is done by using a Neural Autoregressive Flow (NAF) Huang et al. (2018) $f_\psi : \mathcal{Z}^* \to \mathcal{Z}$ parametrized by $\psi \in \Psi$. Neural autoregressive flows are universal density estimation models which are capable of sampling any random variable $Y$ by applying a learned transformation over a base random variable $X$ (Thm. 1 in Huang et al. (2018)).

Given a reconstruction loss $c$ on $\mathcal{X}$, we have the following mean reconstruction loss:

$$\mathrm{RE} := \mathbf{E}_{(x,m)\sim\mathcal{D}}\mathbf{E}_{\hat{x}\sim p(\cdot|z)}\mathbf{E}_{z\sim q(\cdot|x,\phi(x,m))}c(\hat{x},x). \tag{7}$$

We regularize the latent spaces $\mathcal{Z}^*$ and $\mathcal{Z}$ by adding the usual KL term appearing in the VAE Evidence Lower Bound (ELBO) on $\mathcal{Z}^*$:

$$\mathrm{KL}^* := \mathbf{E}_{z^*\sim q^*(.|x)} \log \frac{q^*(z^*|x)}{p^*(z^*)} \tag{8}$$

and an MMD-based regularization on $\mathcal{Z}$ similar the one used in WAEs (see Alg. 2 in Tolstikhin et al. (2017)):

$$\mathrm{MMD}_k\left(q(\cdot|x,\phi(x,m)),p(\cdot)\right) := \left\|\int_{\mathcal{Z}} k(z,\cdot)q(z|x,\phi(x,m)) - \int_{\mathcal{Z}} k(z,\cdot)p(z)\right\|_{\mathcal{H}_k}, \tag{9}$$

where $k : \mathcal{Z}\times\mathcal{Z} \to \mathbf{R}$ is an positive-definite reproducing kernel and $\mathcal{H}_k$ the associated Reproducing Kernel Hilbert Space (RKHS) Berlinet & Thomas-Agnan (2011).

The equations (3), (4) and (5) of Alg. 1 are estimators on a mini-batch of size $n$ of equations (7), (8) and (9) respectively, (5) being the unbiased U-statistic estimator of (9) Gretton et al. (2012).

## 2.2 DISENTANGLING ATTRIBUTES FROM TEMPLATES

Our encoder $q(z|x,\psi)$ thus depends exclusively on $x$ and on the feature space $\Psi$. However, there is no reason, for a random attribute $\psi \in \Psi \neq \phi(x,m)$, that $p(x|z)$ where $z \sim q(z|x,\phi)$ generates variations of the original $x$ with features $\phi$. Indeed, all needed information for reconstructing $x$ is potentially already contained in $z$.

We propose to add an adversarially-learned cost on the latent variable $z_*$ to force the encoder $q_*$ to discard information about the attributes of $x$: Specifically, we train a discriminator neural network $D : \mathcal{Z}^*\times\Psi \to [0,1]$ whose role is to evaluate the probability $D(z^*,\psi)$ that there exists a $(x,m) \in \mathcal{D}$ such that $\psi = \phi(x,m)$ and $z_* \sim q^*(\cdot|x)$. In other words, the aim of the discriminator is to determine if the attributes $\psi$ and the template code $z^*$ originate from the same $(x,m) \in \mathcal{D}$ or if the features $\psi$ are randomly generated. We postpone the explanation on how we sample random features $\psi \in \Psi$ in Sect. 2.3 and suppose for the moment that we have access to a distribution $\nu(\psi)$ over $\Psi$ from which we can sample. The encoder-decoder architecture presented in Sect. 2.1 is trained to fool the discriminator: this means that for a given $(x,m) \in \mathcal{D}$ it tries to produce a template code $z^* \sim q^*(\cdot|x)$ which contains no information about the features $\phi(x,m)$.

In an optimal setting, i.e. when the discriminator is unable to match any $z^* \in \mathcal{Z}^*$ with a particular feature $\psi \in \Psi$, the space of template codes and the space of attributes are decorrelated. All the missing information needed to reconstruct $x$ given $z^* \sim q^*(\cdot|x)$ lies in the transformation $f_{\phi(x,m)}$. Since these transformations between the template space $\mathcal{Z}^*$ and the latent space $\mathcal{Z}$ only depend on the feature space $\Psi$, they tend to be applicable over all template codes $z^*$ and generalize well. During generation time, it is then possible to change the attributes of a sample without changing its template.

The discriminator is trained to maximize

$$\mathcal{L}_{\mathrm{Disc}} := \mathbf{E}_{(x,m)\sim\mathcal{D}}\mathbf{E}_{z^*\sim q^*(\cdot|x)}\left[\mathbf{E}_{\psi\sim\nu(\cdot)}\log D(z^*,\psi) + \log(1-D(z^*,\phi(x,m)))\right]. \tag{10}$$

while the encoder-decoder architecture is trained to minimize

$$\mathcal{R}_{\mathrm{Disc}} := -\mathbf{E}_{(x,m)\sim\mathcal{D}}\mathbf{E}_{z^*\sim q^*(\cdot|x)}\log D(z_i^*,\phi(x,m)). \tag{11}$$

Estimators of Eq. (10) and (11) are given by Eq. (1) and (6) respectively.

## 2.3 PARAMETRIZATION OF THE ATTRIBUTE SPACE

We adopt a particular parametrization of our attribute function $\phi : \mathcal{X} \times \mathcal{M}$ so that we are able to sample *fake* attributes without the need to rely on an existing $(x, m) \in \mathcal{D}$ pair. In the following, we make a distinction between two different cases: the case of continuous *free attributes* and the case of fixed continuous or discrete attributes.

### 2.3.1 FREE ATTRIBUTES

In order to handle free attributes, which denote attributes that are not specified *a priori* but learned. For this, we introduce $d_\Psi$ *attribute vectors* $v_i$ of dimension $d$ together with an *attention module* $\alpha : \mathcal{X} \times \mathcal{M} \to [0, 1]^{d_\Psi}$, where $d_\Psi$ is the intrinsic dimension of the attribute space $\Psi$. By denoting $\alpha_i$ the coordinates of $\alpha$, we then write our attribute function $\phi$ as

$$\phi(x, m) = \sum_{i=1}^{d_\Psi} \alpha_i(x, m) v_i. \tag{12}$$

This approach is similar to the *style tokens* approach presented in Wang et al. (2018). The $v_i$'s are global and do not depend on a particular instance $(x, m)$. By varying the values of the $\alpha_i$'s between $[0, 1]$, we can then span a $d_\Psi$-dimensional hypercube in $\mathbf{R}^d$ which stands for our *attribute space* $\Psi$. It is worth noting that the $v_i$'s are also learned and thus constitute an adaptive basis of the attribute space.

In order to define a probability distribution $\nu$ over $\Psi$ (note that this subspace also varies during training), we are free to choose any distribution $\nu_\alpha$ over $[0, 1]^{d_\Psi}$. We then sample random attributes from $\nu$ by

$$\psi \sim \nu \quad \Longleftrightarrow \quad \psi = \sum_{i=1}^{d_\Psi} \alpha_i v_i \quad \text{where} \quad \alpha_i \sim \nu_\alpha. \tag{13}$$

### 2.3.2 FIXED ATTRIBUTES

We now suppose that the metadata variable $m \in \mathcal{M}$ contains attributes that we want to vary at generation time. For simplicity, we can suppose that this metadata information can be either continuous with values in $[0, 1]^M$ (with a natural order on each dimension) or discrete with values in $[|0, M|]$.

In the continuous case, we write our attribute function

$$\phi(x, m) = \sum_{i=1}^{M} m_i v_i \tag{14}$$

while in the discrete case, we just consider

$$\phi(x, m) = e_m, \tag{15}$$

where $e_m$ is a $d_\Psi$-dimensional embedding of the symbol $m$. It is important to note that even if the attributes are fixed, the $v_i$'s or the embeddings $e_m$ are learned during training.

These two equations define a natural probability distribution $\nu$ over $\Psi$:

$$\psi \sim \nu \quad \Longleftrightarrow \quad \psi = \phi(x, m) \quad \text{where} \quad (x, m) \sim \mathcal{D}. \tag{16}$$

## 3 COMMENTS

We now detail our objective (2) and notably explain our particular choice concerning the regularizations on the latent spaces $\mathcal{Z}^*$ and $\mathcal{Z}$. In Sect. 3.1, we will see that these insights suggest an additional way to "control" the influence of the learned free attributes. In Sect. 3.2, we further discuss about the multiple possibilities that we have concerning the implementation of the attribute function. We list, in Sect. 3.3, the different sampling schemes of VarNet. Finally, Sect. 3.4 is dedicated to implementation details.

## 3.1 Choice of the regularizations on the latent spaces

We discuss our choice concerning the regularizations of the latent spaces and specifically why we chose a KL regularization on $\mathcal{Z}^*$ and an MMD loss on $\mathcal{Z}$.

We found that using a MMD-based regularization on the template space $\mathcal{Z}^*$ resulted in approximated posterior distributions $q^*(\cdot|x)$ with very small variances (almost deterministic mappings). One explanation of this behavior is that the MMD regularization tries to enforce that the aggregated posterior $\frac{1}{N}\sum_{i=1}^{N} q^*(\cdot|x^{(i)})$ matches the prior $p^*$: it does not act on the individual conditional probability distributions $q^*(\cdot|x)$. This degenerate behavior is a side-effect of our adversarial regularization since stochastic encoders have been successfully used in WAEs Rubenstein et al. (2018). When using the the Kullback-Leibler regularization on $\mathcal{Z}^*$, this effect disappear which makes the KL regularization that we considered more suited for VarNet since it helps to keep our model out of a degenerate regime. For some applications, it can still be of interest to have a control over the variance of the conditional probability distributions $q^*(\cdot|x)$. Similar to the approach of Higgins et al. (2016); Burgess et al. (2018), we propose to multiply the KL term by a scalar parameter $\beta > 0$. For $\beta = 1$, we retrieve the original formulation. For $\beta \in ]0, 1[$, decreasing the value of $\beta$ from one to zero decreases the variance of the $q^*(\cdot|x)$. We found no gain in considering values of $\beta$ greater than 1. Examples where this tuning provides an interesting application are given in Sect. A.2.

We now consider the regularization over $\mathcal{Z}$. This regularization is in fact superfluous and could be removed. However, we noticed that adding this MMD regularization helped obtaining better reconstruction losses.

## 3.2 Flexibility in the choice of the attribute function

In this section, we focus on the parametrization of the attribute function $\phi : \mathcal{X} \times \mathcal{Z} \mapsto \mathbf{R}^d$ and propose some useful use cases. The formulation of Sect. 2.3 is in fact too restrictive and considered only one attribute function. It is in fact possible to mix different attributes functions by simply concatenating the resulting vectors. By doing so, we can then combine free and fixed attributes in a natural way but also consider different attention modules $\alpha$. We can indeed use neural networks with different properties similarly to what is done in Chen et al. (2016) but also consider different distributions over the attention vectors $\alpha_i$.

It is important to note that the free attributes presented in Sect. 2.3.1 can only capture *global* attributes, which are attributes that are relevant for *all* elements of the dataset $\mathcal{D}$. In the presence of discrete labels $m$, it can be interesting to consider *label-dependent free attributes*, which are attributes specific to a subset of the dataset. In this case, the attribute function $\phi$ can be written as

$$\phi(x, m) = \sum_{i=1}^{d_\psi} \alpha_i(x, m) e_{m,i}, \tag{17}$$

where $e_{m,i}$ designates the $i^{th}$ attribute vector of the label $m$. With all these possibilities at hand, it is possible to devise numerous applications in which the notions of *template* and *attribute* of an input $x$ may have diverse interpretations.

Our choice of using a discriminator over $\Psi$ instead of, for instance, over the values of $\alpha$ themselves allow to encompass within the same framework discrete and continuous fixed attributes. This makes the combinations of such attributes functions natural.

## 3.3 Sampling schemes

We quickly review the different sampling schemes of VarNet. We believe that this wide range of usages makes VarNet a promising model for a wide range of applications.

We can for instance:

- generate random samples $\hat{x}$ from the estimated dataset distribution:
$$\hat{x} \sim p(\cdot|z) \quad \text{with} \quad z = f_\psi(z^*) \quad \text{where} \quad z^* \sim p^*(\cdot) \quad \text{and} \quad \psi \sim \nu(\cdot), \tag{18}$$
- sample $\hat{x}$ with given attributes $\psi$:
$$\hat{x} \sim p(\cdot|z) \quad \text{with} \quad z = f_\psi(z^*) \quad \text{where} \quad z^* \sim p^*(\cdot), \tag{19}$$

- generate a variations of an input $x$ with attributes $\psi$:

$$\hat{x} \sim p(\cdot|z) \quad \text{with} \quad z = f_\psi(z^*) \quad \text{where} \quad z^* \sim q^*(\cdot|x), \tag{20}$$

- generate random variations of an input $x$:

$$\hat{x} \sim p(\cdot|z) \quad \text{with} \quad z = f_\psi(z^*) \quad \text{where} \quad z^* \sim q^*(\cdot|x) \quad \text{and} \quad \psi \sim \nu(\cdot). \tag{21}$$

Note that for sampling generate random samples $\hat{x}$, we do that by sampling $z^* \sim p^*(\cdot)$ from the prior, $\psi \sim \nu(\cdot)$ from the distribution of the attributes and then decoding $z = f_\psi(z^*)$ decoding it using the decoder $p(\cdot|z)$ instead of just decoding a $z^* \sim p^*(\cdot)$ sampled from the prior. This is due to the fact that, as already mentioned, this MMD regularization is not an essential element of the VarNet architecture: its role is more about fixing the "scale" of the $\mathcal{Z}$ space rather than enforcing that the aggregated posterior distribution exactly matches the prior.

In the case of continuous attributes of the form Eq. (12) or (14), VarNet also provides a new way to navigate in the latent space $\mathcal{Z}$. Indeed, for a given template latent code $z^*$, it is possible to move continuously in the latent space $\mathcal{Z}$ by simply changing continuously the values of the $\alpha_i$ and then considering $z = f_\psi(z^*)$ where $\psi = \sum_{i=1}^{d_\psi} a_i v_i$. The image by the above transformation in the $\mathcal{Z}$ space of the $d_\Psi$ dimensional hypercube $[0,1]^{d_\psi}$ constitutes the *space of variations of the template* $z^*$. Since our feature space bears a measure $\nu$, this space of variations has a probabilistic interpretation. To the best of our knowledge, we think that it is the first time that a meaningful probabilistic interpretation about the displacement in the latent space in terms of attributes is given: We'll see in Appendix A.3 that two similar variations applied on different templates can induce radically different displacements in the latent space $\mathcal{Z}$. We hope that this new technique will be useful in many applications and help go beyond the traditional (but unjustified) linear or spherical interpolations White (2016).

### 3.4 IMPLEMENTATION DETAILS

Our architecture is general and any decoder and encoder networks can be used. We chose to use a NAF[1] for our encoder network. This choice has the advantage of using a more expressive posterior distribution compared to the often-used diagonal Gaussian posterior distributions.

Our priors $p^*$ and $p$ are zero-mean unit-variance Gaussian distributions. For the MMD regularization, we used the parameters used in Tolstikhin et al. (2017) ($\lambda = 10$ and $k(x,y) = C/(C+\|x-y\|_2^2)$ the inverse multiquadratics kernel with $C = 2\dim(\mathcal{Z})$). For the scalar coefficient $\gamma$, we found that a value of 10 worked well on all our experiments.

For the sampling of the $\alpha$ values in the free attributes case, we considered $\nu_\alpha$ to be a uniform distribution over $[0,1]^{d_\psi}$. In the fixed attribute case, we simply obtain a random sample $\{\psi_i\}_{i=1}^n$ by shuffling the already computed batches of $\{\phi(x_i, m_i)\}_{i=1}^n$ (lines 4 and 6 in Alg.1).

## 4 RELATED WORK

The Variation Network generalizes many existing models used for controlled input manipulation by providing a unified probabilistic framework for this task. We now review the related literature and discuss the connections with VarNet.

The problem of controlled input manipulation has been considered in the *Fader networks* paper Lample et al. (2017), where the authors are able to modify in a continuous manner the attributes of an input image. Similar to us, this approach uses an encoder-decoder architecture together with an adversarial loss used to decouple templates and attributes. The major difference with VarNet is that this model has a deterministic encoder which limits the sampling possibilities as discussed in Sect. A.2. Also, this approach can only deal with fixed attributes while VarNet is able to also learn meaningful free attributes. In fact, VAEs Kingma & Welling (2013), WAEs Tolstikhin et al. (2017); Rubenstein et al. (2018) and Fader networks can be seen as special cases of VarNet.

Recently, the *Style Tokens* paper Wang et al. (2018) proposed a solution to learn relevant free attributes in the context of text-to-speech. The similarities with our approach is that the authors condition an encoder model on an adaptive basis of style tokens (what we called attribute space in this

---

[1]We used the implementation of Huang et al. (2018) available at https://github.com/CW-Huang/NAF

work). VarNet borrows this idea but cast it in a probabilistic framework, where a distribution over the attribute space is imposed and where the encoder is stochastic. Our approach also allows to take into account fixed attributes, which we saw can help shaping the free attributes.

Traditional ways to explore the latent space of VAEs is by doing linear (or spherical White (2016)) interpolations between two points. However, there are two major caveats in this approach: the requirement of always needing two points in order to explore the latent space is cumbersome and the interpolation scheme is arbitrary and bears no probabilistic interpretation. Concerning the first point, a common approach is to find, a posteriori, directions in the latent space that accounts for a particular change of the (fixed) attributes Upchurch et al. (2016). These directions are then used to move in the latent space. Similarly, Hadjeres et al. (2017) proposes a model where these directions of interest are given a priori. Concerning the second point, Laine (2018) proposes to compute interpolation paths minimizing some energy functional which result in interpolation curves rather than interpolation straight lines. However, this interpolation scheme is computationally demanding since an optimization problem must be solved for each point of the interpolation path.

Another trend in controlled input manipulation is to make a posteriori analysis on a trained generative model Engel et al. (2017); Adel et al. (2018); Upchurch et al. (2016); Cao et al. (2018) using different means. One possible advantage of these methods compared to ours is that different attribute manipulations can be devised after the training of the generative model. But, these procedures are still costly and so provide any real-time applications where a user could provide on-the-fly the attributes they would like to modify. One of these approaches Cao et al. (2018) consists in using the trained decoder to obtained a mapping $\mathcal{Z} \mapsto \mathcal{X}$ and then performing gradient descent on an objective which accounts for the constraints or change of the attributes. Another related approach proposed in Engel et al. (2017) consists in training a Generative Adversarial Network which learns to move in the vicinity of a given point in the latent space so that the decoded output enforces some constraints. The major difference of these two approaches with our work is that these movements are done in a unique latent space, while in our case we consider separate latent spaces. But more importantly, these approaches implicitly consider that the variation of interest lies in a neighborhood of the provided input. In Adel et al. (2018) the authors introduce an additional latent space called *interpretable lens* used to interpret the latent space of a generative model. This space shares similarity with our latent space $\mathcal{Z}^*$ and they also propose a joint optimization for their model, where the encoder-decoder architecture and the interpretable lens are learned jointly. The difference with our approach is that the authors optimize an "interpretability" loss which requires labels and still need to perform a posteriori analysis to find relevant directions in the latent space.

## 5 CONCLUSION AN FUTURE WORK

We presented the Variation Network, a generative model able to vary attributes of a given input. The novelty is that these attributes can be fixed or learned and have a sound probabilistic interpretation. Many sampling schemes have been presented together with a detailed discussion and examples. We hope that the flexibility in the design of the attribute function and the simplicity, from an implementation point of view, in transforming existing encoder-decoder architectures (it suffices to provide the encoder and decoder networks) will be of interest in many applications.

For future work, we would like to extend our approach in two different ways: being able to deal with partially-given fixed attributes and handling discrete free attributes. We also want to investigate the of use stochastic attribute functions $\phi$. Indeed, it appeared to us that using deterministic attribute functions was crucial and we would like to go deeper in the understanding of the interplay between all VarNet components.

## ACKNOWLEDGMENTS

Omitted for double blind review.

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

## A  EXPERIMENTS

We now apply VarNet on MNIST in order to illustrate the different sampling schemes presented in Sect. A.

In all these experiments, we choose to use a simple MLP with one hidden layer of size 400 for the encoder and decoder networks. We present and comment results for different attribute functions and different sampling schemes. The different attribute functions we considered are

- 1Free: one-dimensional free attribute space (Eq. (12) with $d_\Psi = 1$),
- 2Free: two-dimensional free attribute space (Eq. (12) with $d_\Psi = 2$),
- 1Free+1FixedLabel: one-dimensional free attribute space (Eq. (12) with $d_\Psi = 1$) concatenated with a fixed attribute which uses the labels of the digits (Eq. (15) with $M = 10$),
- 1Free+1FreeLabel: one-dimensional free attribute space (Eq. (12) with $d_\Psi = 1$) concatenated with a label-dependent free attribute of dimension 1 (Eq. (17) with $M = 10$ and $d_\psi = 1$).

### A.1  UNCONSTRAINED AND CONSTRAINED SAMPLING SCHEMES

We display in Figure 2 samples obtained with the sampling procedures Eq. (18) and Eq. (19) when considering the 1Free+1FixedLabel attribute function. The results are in par with other probabilistic generative models on this task like VAEs, CVAEs or WAEs.

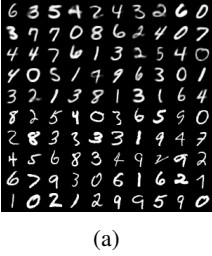  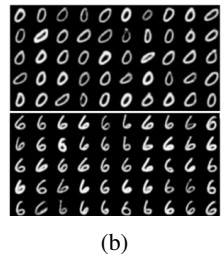  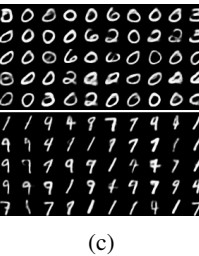

(a)  (b)  (c)

Figure 2: Different sampling schemes. Fig. 2a: sampling from the dataset distribution using Eq. (18) using the 1Free+1FixedLabel attribute function. Fig. 2b: sampling elements with fixed attribute $\psi$ using Eq. (19) with the 1Free+1FixedLabel attribute function. Fig. 2c: same as Fig. 2b but using the 1Free attribute function. In Fig 2c and 2b, two sets of samples (top and bottom) corresponding to two different values of $\psi$ are shown.

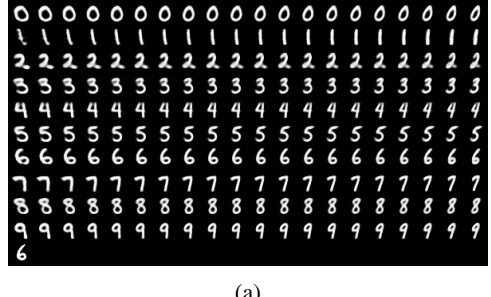  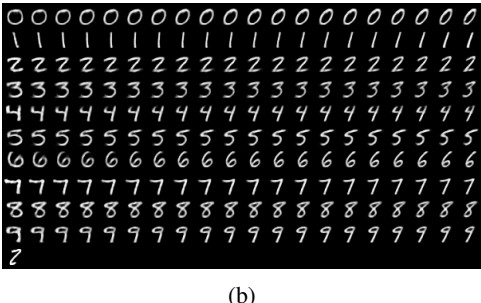

(a)  (b)

Figure 3: Visualization of the spanned *space of variations* using two different inputs (shown in the last row). The attribute function 1Free+1FixedLabel is used. The values of $\alpha_i$ for the free attributes (see Eq. (12)) increase linearly from 0 to 1.

### A.2  UNDERSTANDING FREE ATTRIBUTES

From Fig. 2b, we see that the fixed label attribute have clearly been taken into account, but it can be hard to grasp which high-level attribute the free attribute function has captured. In order to visualize


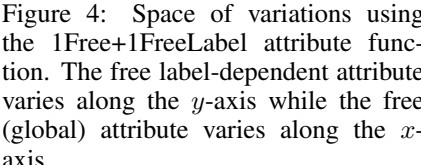

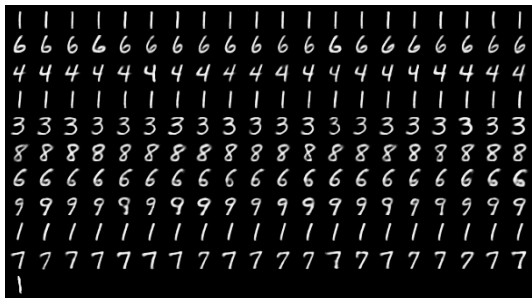

Figure 4: Space of variations using the 1Free+1FreeLabel attribute function. The free label-dependent attribute varies along the $y$-axis while the free (global) attribute varies along the $x$-axis.

Figure 5: Sampling scheme Eq. (21) using the 1Free+1Label attribute function. Each row shows samples obtained by sampling $z_* \sim q_*(.|x)$ for a fixed random feature $\psi$. The original input is shown on the last line.

this, we plot in Fig. 3 a visualization of the *space of variations* spanned by a given template latent code $z^*$. From these plots, it appears that the attribute vector encodes a notion of rotation meaningful for this digit dataset and it is interesting to note how different templates produce different "writing styles". Free attributes can thus be particularly interesting for capturing high-level features, such like rotation, that cannot be described in an absolute way or which are ill-defined.

By observing carefully Fig. 3, we note that the variations generated by varying the free attribute applies to all digit classes, irrespective of their label. In such a case, it is impossible to obtain different "writing conventions" for the same digit (like cursive/printscript style for the digit "2") by only modifying the attributes. We show in Fig. 4 that, by considering free label-dependent attributes, we are able to smoothly go from one "writing convention" to the other one.

We can gain further insight about the notion of template and attribute using the sampling scheme of Eq. (21). This sampling exploits the stochasticity of the encoder $q^*(\cdot|x)$ in order to generate variations of a given input $x$ using a fixed attribute $\psi$. An example of such variations is given in Fig. 5. The underlying idea is that, even for a given attribute $\psi$, there are multiple ways to generate variations of $x$ with attributes $\psi$. We believe that this stochasticity is essential since, in many applications, there should not exist only one way to make variations.

The parametrization of the attribute function has a crucial effect on the high-level features that they will able to capture. For instance, if we do not provide any label information, the information present in the template and the information contained in the attribute function can differ drastically. Figure 6 show different space of variations where no label information is provided. The concepts captured in these cases are then related to thinness/roundness. Our intuition is that the free attributes capture the most general attributes of the dataset.

For some applications, variation spaces such as the one displayed in Fig. 6a, 6b or 6d are not desirable because they may tend to move too "far away" from the original input. As discussed in Sect. 3.1, it is possible to reduce how "spread" the spaces of variation are by modifying the $\beta$ parameter multiplying the KL term in the objective Eq. (2). An example of such a variation space is displayed in Fig. 6c.

From all examples above, we see that our architecture is indeed capable of decoupling templates from learned attributes and that we have two ways of controlling the free attributes that are learned: by modifying the KL term in the objective Eq. (2) and by carefully devising the attribute function. Indeed, the learned free attributes can capture different high-level features depending on the other fixed attributes they are coupled with.

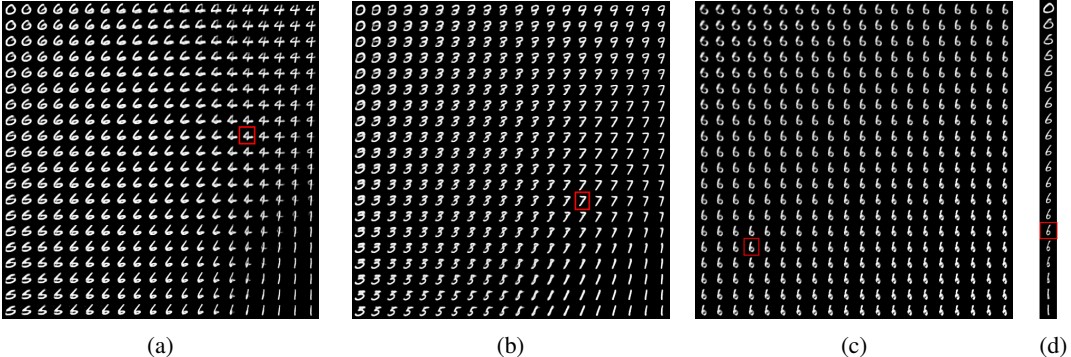

Figure 6: Figures 6a, 6b and 6c display the space of variations using the 2Free attribute function for two different input. Fig. 6d display the space of variations using the 1Free attribute function. Figure 6c was generated using a model trained with a low KL penalty ($\beta = 0.1$)

### A.3 Moving in the latent space: beyond interpolations

VarNet proposes a novel solution to explore the latent spaces. Usual techniques to navigate in the space of VAEs such as interpolations or the use of attribute vectors (distinct from what we called attribute vectors in this work) are mostly intrinsically-based on moving using straight lines. This assumes that the underlying geometry is euclidean, which is not the case, and forgets about the probabilistic framework. Also, computing attribute vectors requires data with binary labels which are not always available.

On the contrary, our approach grants a sound probabilistic interpretation of the attributes and the variations they generate. Indeed, when the discriminator is fooled by the encoder-decoder architecture, the attributes are distributed according to $\nu$ which has a simple interpretation (it is the push-forward of the $\nu_\alpha$ distribution which is considered to be a uniform distribution in all these examples). Also, thinking about variations as a subspace of smaller dimension than the whole latent space makes much sense for us.

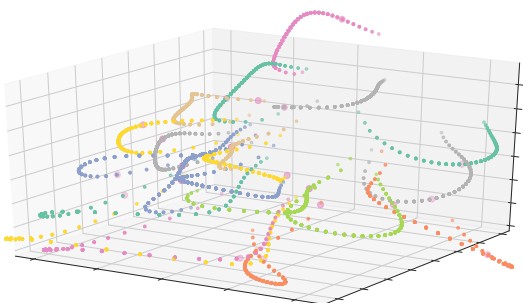

Figure 7: Variation spaces (shown in $\mathcal{Z}$) of a VarNet trained using the 1Free attribute function, for different $z^*$. We plotted $\{z = f_\psi(z^*)\}$ for $\psi = \alpha_1 v_1$ where $\alpha_1 \in [0.0, 0.05, \ldots, 0.95, 1.0]$ and random $z^*$. Visualized using a 3D PCA in $\mathcal{Z}$.

Figure 7 shows a visualization in the latent space $\mathcal{Z}$ of the variation spaces spanned by moving with constant steps in the attribute space $\Psi$. Two key elements appear: constant steps in the attribute space do not induce constant steps in the $\mathcal{Z}$ space and variation spaces are extremely diverse (they are not translated versions of a unique variation space). For us, this advocates for the fact that displacements in the latent spaces using straight lines have a priori no meaningful interpretation: the same change of attributes for two different inputs can lead to radically different displacements in the latent space. More generally, our proposition of parametrizing attribute-related displacements in a latent space using flows conditioned on a simpler space is appealing from a conceptual point of view since we do not mix, in the same latent space, its probabilistic interpretation given by the prior and its ability to grant meaningful ways to vary attributes.

