# OpenReview forum: "Variation Network: Learning High-level Attributes for Controlled Input Manipulation"
_ICLR.cc/2019/Conference_

### Official Review · AnonReviewer2 · 2018-11-05
**Lack of clarity and almost no experiments**

**Rating:** 4
**Confidence:** 3

**Review:**

This paper proposes a generalization of variational auto-encoders to account for meta-data (attributes), learning new ones, in a way that these can be controlled to generate new samples. The model learns how to decouple the attributes in an adversarial way by means of a discriminator. The problem is interesting, but I found two main issues with this paper:
1.- Lack of clarity: I found the paper difficult to follow, even after reading Sec. 2 and 3 several times.
2.- Almost absence of experiments: The paper only has one experiment, which is in the appendix, and is about sampling using the MNIST dataset. Given that this paper proposes a model, whose properties can be assessed by means of experiments, the fact that there is nothing of the kind provides no support to any benefits the model may have.

Other points:
What in the model prevents the solution of z_* being just random (independently of x)?

This paper seems relevant Esser, Patrick, Ekaterina Sutter, and Björn Ommer. "A Variational U-Net for Conditional Appearance and Shape Generation." Proceedings of the IEEE Conference on Computer Vision and Pattern Recognition. 2018.

---

### Official Review · AnonReviewer1 · 2018-11-05
**Interesting Work**

**Rating:** 6
**Confidence:** 2

**Review:**

The paper proposes a generative network capable of generating variations of a given input, conditioned on an attribute. Earlier papers generated variations of the input in the presence of the attribute and this attribute was assumed to be known during training. This paper proposes to automatically discover these attribute and thus work to produce variations even in the absence of known attribute information.

The paper is dense, but it is well written. It has mixed ideas from several papers - the basic VAE architecture, combined with a discriminator and regularizations over latent space. The key thing, of course, is the design of the attribute function. There seems to be an interesting interaction between the encoder, discriminator and the attribute function that requires more investigation. This is acknowledged in the conclusion as well.

The work is original and the results on the MNIST dataset are very interesting. I think the significance of this work lies in the fact that this can be a starting point for several interesting future works in this direction.

---

### Official Review · AnonReviewer3 · 2018-11-11
**Paper lacking an experimental section**

**Rating:** 3
**Confidence:** 4

**Review:**

This paper introduces a new framework for learning an interpretable representation of images and their attributes. The authors suggest decomposing the representation into a set of 'template' latent features, and a set of attribute-based features. The attribute-based features can be either 'free', i.e. discovered from the data, or 'fixed', i.e. based on the ground truth attributes. The authors encourage the decomposition of the latent space into the 'template' and the 'attributes' features by training a discriminator network to predict whether the attributes and the template features come from the same image or not.

While the idea is interesting, the paper is lacking an experimental section, so the methodology is impossible to evaluate. Furthermore, while the authors spend many pages describing their methodology, the writing is often hard to follow, so I am still confused about the exact implementation of the attribute features \phi(x, m) for example. The authors do point to the Appendix for their Experiments section, however this is not a good idea. The paper should be self-contained and the authors should not assume that their readers will read the information presented in the Appendix, which is always optional.

Unfortunately, even the experimental section presented in the Appendix is not comprehensive enough to evaluate the proposed method. The authors train the model on a single dataset (MNIST), no baseline or ablation results are presented, and all the results are purely qualitative. Given that the ground truth attribute decomposition for MNIST is not known, even the qualitative results are impossible to evaluate. I recommend that the authors present quantitive results in the updated version of their paper (i.e. disentanglement metric scores, the log-likelihood of the reconstructions), including new experiments on a dataset like dSprites or CelebA, where the ground truth attributes are known.

---

### Author Response · Authors · 2018-11-26
**Reply**

We first thank the reviewers for their reviews and now answer individually to the concerns raised.

-Review#1: Thanks a lot. We are glad that the interest of our contributions has been understood and appreciated. On top of the contributions mentioned, we also would like to mention the sound probabilistic formulation of our model which we think can be valuable since the probabilistic interpretation of the attributes allows to perform meaningful interpolations and to introduce a new sampling procedure.
Indeed, understanding the interaction and the dynamics between all the elements highly interests us and we will work on that in the future: Especially, we would like to investigate how the dimensionality of the attribute space affects the learnt attributes and how we can shape these attributes by providing, for instance, weaker attribute functions.


-Review#2: We are sorry that you find our paper hard to follow. Can you be more specific or provide us with ideas for improvement? We are quite surprised about this statement since we tried to be as detailed as possible: all aspects of the model are discussed, motivated and progressively introduced; we also provided a detailed algorithm together with a figure of our architecture.

The experimental part illustrates the different and novel sampling schemes offered by VarNet on a well-known and simple dataset. It is here for illustration purposes and it is not intended to prove anything. Our paper is not about a specific model or implementation. We chose MNIST because it allows to easily understand what this framework provides, without the need to focus on the implementation of the encoder, decoder and attribute function. That's why we chose simple MLPs for the encoder, decoder and attribute function and put this experimental part in appendix.

 The paper you propose is indeed relevant and we will include it in the related works. But this approach, like the Fader networks, requires known attributes (appearance).

 Concerning the last question, what prevents z* to be independent of x is that the attribute space is of low dimensionality (as in the Style Tokens paper), so you cannot fully reconstruct x by only considering its attributes. In the degenerate case, z* is a noise independent of x and VarNet amounts to a WAE.


-Review#3: See reply to reviewer#2 concerning the clarity of our paper or the discussion concerning the experimental part. Indeed, we believe that the part in appendix is optional and is just here for illustrative purposes.

 \phi(x, m) is just any neural network "This attribute function is a deterministic neural network that will be learned during training and whose aim is to compute attributes of x " Sect. 2.1. In the experiments, it is just a MLP (depending on x only or on x and m). We will precise that.

 Concerning your 3rd paragraph, we would like to stress upon the fact that there is no ground truth for the attributes. The purpose of this framework is not about that nor about disentanglement. Also, this is a framework to devise generative models with novel sampling properties, not about a specific implementation, so there is no point in showing log-likelihood of the reconstructions ( even if we take the same encoder and decoder networks, how to fairly compare this with a VAE?).
 When applied bluntly on CelebA and by considering the "Eyeglasses" attribute, we are for instance capable of adding
 different style of glasses on the same face simply by sampling. The attribute function learns in this case some kind of color palette. On Dsprites, we can obtain two-dimensional planes of variations accounting for the scale and y-position (controlled by the x-axis) and the rotation and x-position (controlled by the y-axis). There is indeed no reason why we could obtain a dimension accounting for only one attribute. We also applied this framework to the generation of sequences of discrete symbols and on sound generation with successful results. Since these experiments require more tuning about the
 encoder, decoder and attribute functions, we preferred to only display the simplest experiment allowing to understand and focus on the possibilities offered by VarNet.

---

> ### Comment · AnonReviewer2 · 2018-12-13
> **Trying to make the paper easier to follow**
>
> Here there are some specifics about why I found the paper difficult to follow.
>
> There are isolated statements that lack a motivation that can guide the reader about why this was a logical step to do. Two examples:
> "The idea is then to compute z from z∗ by applying a transformation parametrized only by the feature space Ψ" --> what is the motivation?
> "We now consider the regularization over Z. This regularization is in fact superfluous and could be removed." --> why is that?
>
> The is a lack of justification in many places in which many things are taken for granted, or there is not a clear cause-effect flow. To mention three examples:
> - "However, there is no reason, for a random attribute ψ ∈ Ψ /= φ(x, m), that p(x|z) where z ∼ q(z|x, φ) generates variations of the original x with features φ" --> what is the role of ψ given that it is not mentioned in the rest of the sentence?
> - Regarding 2.3.1, it is not clear why the distribution ν_α turns out to be similar to the outputs of α_i (even more so if the discriminator encourages them to be turned apart, and so making it easy to separate z* and ψ, while allowing z* to contain all information about x).
> - In sec. 3.1. "This degenerate behavior is a side-effect of our adversarial regularization since stochastic encoders have been successfully used in WAEs Rubenstein et al. (2018)" it is not clear what adversarial regularization has to do with the degenerate behavior, and how the reference gives any support to that claim.
>
> Overall, given that the proposed model has a fair degree of complexity and thus is difficult to explain, it may be helpful to illustrate and motivate its parts with a specific example (e.g. an image of a particular thing), describing for each element of the approach, the kind of information it is supposed to contain for that particular case.

---

### Meta-Review · Area_Chair1 · 2018-12-15

**Confidence:** 5
**Recommendation:** Reject

**Metareview:**

The authors propose a generative model based on variational autoencoders that provides means to manipulate the high-level attributes of a given input. The attributes can be either pre-defined ground truth attributes or unknown attributes automatically discovered from the data.

While the reviewers acknowledged the potential usefulness of the proposed approach, they raised important concerns that were viewed by AC as a critical issue: (1) very limited experimental evaluation (e.g. no baseline or ablation results, no quantitative results); comparisons on other more complex datasets and more in-depth analysis would substantially strengthen the evaluation and would allow to assess the scope of the contribution of this work  – see, for example, R3’s suggestion to use other dataset like dSprites or CelebA, where the ground truth attributes are known; (2) lack of presentation clarity – see R2’s latest comment how to improve.

A general consensus among reviewers and AC suggests, in its current state the manuscript is not ready for a publication. It needs clarification, more empirical studies and polish to achieve the desired goal.